# Genetic Ablation of LAT1 Inhibits Growth of Liver Cancer Cells and Downregulates mTORC1 Signaling

**DOI:** 10.3390/ijms24119171

**Published:** 2023-05-24

**Authors:** Sun-Yee Kim, Qunxiang Ong, Yilie Liao, Zhaobing Ding, Alicia Qian Ler Tan, Ler Ting Rachel Lim, Hui Min Tan, Siew Lan Lim, Qian Yi Lee, Weiping Han

**Affiliations:** 1Institute of Molecular and Cell Biology, Agency for Science, Technology and Research (A*STAR), 11 Biopolis Way, #02-02, Helios, Singapore 138667, Singapore; kim_sunyee@imcb.a-star.edu.sg (S.-Y.K.); ongqx@imcb.a-star.edu.sg (Q.O.); ding_zhaobing@imcb.a-star.edu.sg (Z.D.); alicia_tan@imcb.a-star.edu.sg (A.Q.L.T.); rachel_lim@imcb.a-star.edu.sg (L.T.R.L.); tan_hui_min@imcb.a-star.edu.sg (H.M.T.); qianyi.lee@cargene.com (Q.Y.L.); 2Duke-NUS Medical School, National University of Singapore, Singapore 169857, Singapore

**Keywords:** LAT1, liver cancer, amino acid transport, mTORC, branched chain amino acids, *slc7a5*

## Abstract

Through a comprehensive analysis of the gene expression and dependency in HCC patients and cell lines, LAT1 was identified as the top amino acid transporter candidate supporting HCC tumorigenesis. To assess the suitability of LAT1 as a HCC therapeutic target, we used CRISPR/Cas9 to knockout (KO) LAT1 in the epithelial HCC cell line, Huh7. Knockout of LAT1 diminished its branched chain amino acid (BCAA) transport activity and significantly reduced cell proliferation in Huh7. Consistent with in vitro studies, LAT1 ablation led to suppression of tumor growth in a xenograft model. To elucidate the mechanism underlying the observed inhibition of cell proliferation upon LAT1 KO, we performed RNA-sequencing analysis and investigated the changes in the mTORC1 signaling pathway. LAT1 ablation resulted in a notable reduction in phosphorylation of p70S6K, a downstream target of mTORC1, as well as its substrate S6RP. This reduced cell proliferation and mTORC1 activity were rescued when LAT1 was overexpressed. These findings imply an essential role of LAT1 for maintenance of tumor cell growth and additional therapeutic angles against liver cancer.

## 1. Introduction

Liver cancer is the third most fatal cancer, accounting for 8.3% of total deaths in 2020 (World Health Organization, Geneva, Switzerland) [1]. Hepatocellular carcinoma (HCC) is a prevalent type of liver cancer in adults, and there is an urgent need to identify more effective therapeutic targets for patients in advanced stages [2]. While immunotherapy has become a clinically validated treatment for many cancers, it has shown only limited efficacy for HCC patients [3].

Emerging evidence suggests that targeting cancer metabolism could be a promising avenue for therapeutic interventions [4]. Given that amino acids, and not glucose, account for the majority of metabolite mass in proliferating cancer cells, there is increasing interest on amino acid metabolism in liver cancer [5,6,7,8]. For instance, glutamine uptake is increased in cancer cells as it is a major source of carbon and nitrogen to synthesize amino acids, nucleic acids, and lipids [6]. Serine and glycine are also sources of carbon in cancer cells and restricted uptake of either amino acid can inhibit tumor growth and extend the overall survival of tumor-bearing mice [7]. Arginine stimulates protein synthesis through the activation of mTOR signaling and promotes cell proliferation in cancer cells [9]. Branched-chain amino acids (BCAAs)—valine, leucine, and isoleucine—are also taken up from the microenvironment by tumors, serving as essential amino acids for cancer growth and participating in various biosynthetic pathways [10,11]. Moreover, BCAA metabolism has been shown to play functional roles in the progression of different cancer types, although its specific functions in HCC remain unclear [12,13,14].

The roles of numerous amino acid transporters in cancer cells have been examined by several studies [15]. Particularly, L-type amino acid transporter 1 (LAT1) has a major role in BCAA uptake by hepatocytes [16] and is also upregulated in a wide spectrum of cancers to increase protein synthesis and promote cell growth [17,18,19]. Targeted inhibition of LAT1 with shRNAs causes detrimental effects on intracellular BCAA levels, cell proliferation, and cell cycle progression in various human cancers [20,21,22,23]. In additional, pharmacological intervention with specific LAT1 inhibitors [24,25] causes significant inhibition of BCAA uptake and cell growth in human colon cancer [21,26], osteosarcoma [27], prostate cancer [28], and lymphoblastoid cells [20]. However, these methods suffer from incomplete knockdown and a lack of specificity, which could lead to misleading results. Thus, there is a clear gap to ascertain how LAT1 and BCAA import affects liver cancer cell growth.

Our previous study has identified a significant loss of BCAA catabolism and a corresponding accumulation of intracellular BCAAs in HCC tumors. Moreover, decreasing the amount of intracellular BCAAs by either reducing or removing dietary BCAA intake or increasing BCAA catabolism significantly suppressed HCC cell proliferation in vitro and improve overall survival in vivo. [14] BCAAs, particularly leucine, also activate the mTORC1 pathway that leads to cell proliferation. Cellular uptake of BCAAs limits the availability of BCAAs in cells. As LAT1 is a prime transporter of BCAAs, we therefore hypothesize that modulation of LAT1 in cancer cells can be a more specific and effective method for controlling intracellular BCAA levels and cancer progression compared to dietary control.

## 2. Results

### 2.1. Expression of LAT1 Is Associated with Clinical Outcomes of HCC Patients

We hypothesize that reducing intracellular BCAA levels by ablation of the relevant dominant transporter is crucial for limiting cancer progression. Both L-type and B^0^ type of solute carrier (SLC) transporters are responsible for transporting BCAAs into hepatocytes. Thus, we first investigated the L-type and B^0^-type SLC expression in individual samples from the Cancer Genome Atlas (TCGA) data. We found that L-type amino acid transporters (LATs) are upregulated in most tumor samples compared to adjacent non-tumor samples. LAT1 (encoded by *slc7a5* gene) was most highly upregulated in tumor samples among the four LATs (Figure 1a). These findings led us to further investigate the changes in LAT1 expression in HCC samples from TCGA data. The gene expression profiling data from 371 HCC patient and 50 normal samples were used to re-analyze the mRNA level of *slc7a5*. The data clearly shows the difference in gene expression distribution between two groups, with the expression of *slc7a5* in cancer tissue higher than in normal tissue (Figure 1b). In addition, LAT1 protein expression profiling data from 165 HCC patient and 165 normal samples consistently indicated that protein level of LAT1 was increased in HCC (Figure 1b). Thus, overexpression of LAT1 mRNA and protein could be associated with HCC tumor progression. To determine the prognostic significance of LAT1 expression in HCC, we next performed a survival analysis using the TCGA-HCC dataset. Disease-specific survival (DSS) analysis, as represented in the Kaplan-Meier plot, demonstrated that higher expression of LAT1 was associated with a significantly worse prognosis for HCC patients (Figure 1c). Specifically, the hazard ratio of HCC-related death was significantly increased in patients with medium-high grade HCC tumors (Grade 2 and 3) possessing a higher LAT1 expression level (Figure 1d). We further confirmed the RNA and protein expressions of LATs by qRT-PCR and Western blot in the Diethylnitrosamine (DEN) mice model. We observed a significant increase in mRNA and protein expression of LAT1 in tumor liver tissues compared to adjacent non-tumor tissues (Figure 1e and Appendix A). Taken together, all the clinical data and mouse model results suggest a strong association between LAT1-driven BCAA transport and HCC development/progression and implies LAT1 is a potential target for liver cancer treatment.

### 2.2. CRISPR/Cas9 Effectively Knocks out LAT1 Expression, Reduces BCAA Transport, and Suppresses Cell Proliferation In Vitro and in Mouse Xenograft Model

To further explore the functional significance of amino acids, especially BCAAs in HCC tumorigenesis, and validate the phenotypic relevance of different LAT isoforms, we performed an unbiased screening for all the amino acid transporters based on DepMap database (Figure 2a). Remarkably, the dependency score in CRISPR knockout (KO) screening showed that KO of LAT1 has the strongest impairment on HCC cell line proliferation and survival among all the amino acid transporters. Interestingly, LAT1 is the only member of the LAT family to result in a significant effect on HCC survival. We had attempted to perform shRNA LAT1 on Hep3B, HepG2 and Huh7 cells, but the knockdown efficiency was not complete (Appendix A). Therefore, we decided to create LAT1 KO in the HCC cell line using CRISPR/Cas9 to introduce permanent and dysfunctional mutation. HCC cell lines were each transfected with CRISPR/Cas9 vectors encoding three unique sgRNA sequences against the *slc7a5* gene. Then, cells were subjected to FACS isolation of cells with high levels of GFP encoded by pSpCas9-2A-GFP. Finally, one clone in each cell line for LAT1 KO (LAT1^KO^) and LAT1 wildtype control (LAT1^WT^) were selected for this study. PCR analysis, sanger sequencing around the sgRNA targeted site (Appendix A), and examination of LAT1 protein expression demonstrated successful KOs of LAT1 protein, but not LAT2, LAT3, and LAT4 in HCC cell lines (Figure 2b and Appendix A). We next investigated the impact of LAT1 ablation on its amino acid transport activity by measuring BCAA content in the cells. LAT1^KO^ demonstrated a 40% reduction in BCAA levels compared to LAT1^WT^ (Figure 2c). We thereafter investigated the effect of LAT1 KO on in vitro HCC cell proliferation. LAT1 ablation dramatically decreased cell proliferation as demonstrated by real-time cell proliferation analysis, SRB assay and BrdU incorporation assay in Huh7 cell line compared to LAT1^WT^ (Figure 2d–f). To further examine the effect of LAT1 on tumor growth in vivo, we used a immunodeficient mouse xenograft tumor model with subcutaneously implanted Huh7 cells. Exponentially growing LAT1^KO^ or LAT1^WT^ cells were injected into the flank of immunodeficient mice (*n* = 6), and tumor growth was continually monitored with calliper measurement. Consistent with in vitro studies, lack of LAT1 led to significant suppression of tumor growth compared with wildtype control in xenograft model (Figure 2g,h). After three weeks of injection, the mean tumor volume of KO cells injected mice was 620.75 ± 166.66 mm^3^, which was significantly smaller than that in wildtype control cells (1683.52 ± 271.95 mm^3^). To clarify the in vivo tumor growth suppression mechanism presented in LAT1^KO^ injected mice, we conducted immunohistochemistry with Ki67 cell proliferation marker. Histology of the LAT1^KO^ and LAT1^WT^ xenografts was similar, but we observed a reduction of Ki67 positive cells in the LAT1^KO^ injected group (Figure 2i). These findings imply an essential role of LAT1 for maintenance of tumor cell amino acid homeostasis and proliferation.

### 2.3. Lentiviral-LAT1 Infection Increases Cellular BCAA Levels and Rescues Cell Proliferation to Similar Levels as Cell Cultured in Wildtype Control Huh7 Cells

We next asked if LAT1 restoration could rescue cell survival and amino acid content in LAT1^KO^. To stably express LAT1 into cells, lentivirus-LAT1 (pLenti-LAT1) or corresponding empty vector (EV) were transduced into LAT1^KO^ or LAT1^WT^ cells. When LAT1 stably re-expressed into LAT1^KO^ by lentivirus transduction, the LAT1 protein level became similar to that in LAT1^WT^ transduced by EV (Figure 3a). We next investigated the recovery of LAT1 amino acid transport activity by measuring intracellular BCAA content in both group of cells transduced with EV or pLenti-LAT1. Re-expression of LAT1 into LAT1^KO^ caused intracellular BCAA content to increase to 90% compared to LAT1^WT^ transduced with EV (Figure 3b). Real-time cell proliferation assay and SRB assay suggested that LAT1 rescue completely restores the growth of LAT1^KO^, as revealed by two-way ANOVA analysis (Figure 3c,d). Collectively, these results suggested that ablation of LAT1 suppresses HCC proliferation by reducing intracellular amino acid content, but these effects could be rescued by stably re-expression of LAT1.

### 2.4. LAT1 Supports Proliferation of HCC by mTORC1 Pathway

To further elucidate the mechanism underlying the inhibition of cell proliferation in LAT1^KO^, we performed RNA-sequencing (RNA-seq) analysis for both LAT1^WT^ and LAT1^KO^ samples. We characterize the relationship between groups by a Principal Component Analysis (PCA). The processed PCA data revealed distinct clustering patterns within each sample group (Figure 4a), prompting us to further investigate gene expression patterns by performing clustering analysis. Subsequently, we identified a total of 2179 differentially expressed genes (DEGs), with 1228 genes being upregulated and 951 genes downregulated in cultured LAT1^KO^ cells (Figure 4b).

To gain insight into the enriched biological processes and pathways associated with LAT1^KO^, we conducted pathway analysis using Ingenuity Pathway Analysis (IPA) on genes with a False Discovery Rate (FDR) < 0.05. Notably, we found that genes in the Mammalian target of rapamycin complex 1 (mTORC1) signaling, 70-kDa ribosomal protein S6 kinase (p70S6K) signaling, and eukaryotic initiation factor 2 (eIF2) and eukaryotic initiation factor 4 (eIF4) signaling were significantly over-represented in LAT1^KO^ (Figure 4c), supporting our observations that LAT1^KO^ suppresses cell proliferation. To further explore the specific gene sets in enrichment pathways, we performed KEGG analysis for DEGs in the cluster of mTOR signaling and translation initiation factor. Most of the DEGs in these 2 pathways were downregulated (Figure 4d), collaboratively suggesting that LAT1 KO caused an overall suppression on mTORC1 signaling and protein translation.

The mTORC1 pathway plays a key role in cell growth in HCC and amino acids are essential signaling molecules to activate mTORC1 [29,30]. To further understand the significance of LAT1 in this response, we assessed the expression or phosphorylation of the direct mTORC1 substrates of p70S6K, eukaryotic initiation factor 4E-binding protein 1 (4EBP1), and unc51-like kinase 1 (ULK1). In the LAT1^KO^ clone, there was a markedly reduced expression and phosphorylation of p70S6K compared to Huh7 cells or LAT1^WT^, while expression and phosphorylation of 4EBP1 and ULK1 appeared unchanged (Figure 4e). We also measured phosphorylation on ribosomal protein S6 (S6RP), which is phosphorylated by p70S6K in a mTORC1 dependent manner and thus indicates mTORC1 activity. LAT1^KO^ markedly reduced the phosphorylation of S6RP (Figure 4f). Since p70S6K is a regulator of translational initiation, we further assessed translational initiation factors activity. Phosphorylation of eIF4b was decreased and that of eIF2a was increased (Figure 4g), suggesting the stimulation of translational initiation was inhibited in LAT1^KO^. These reduced activities of p70S6K and S6RP were rescued when LAT1 was re-expressed by pLenti-LAT1 transduction (Figure 4h). Collectively, these observations imply that LAT1-mediated amino acid transport is required to activate translation initiation in HCC.

## 3. Discussion

Cancer cells exhibit Warburg effect, where they fundamentally alter their metabolism to decrease dependence on aerobic glycolysis [31]. Cancer cells convert pyruvate into lactate instead of directing the former towards the TCA cycle and oxidative phosphorylation. This metabolic reprogramming enables cancer cells to generate precursors for the formation of new biomass. To support sustained biomass accumulation and cell division, cancer cells require amino acids from the tumor microenvironment. Non-essential amino acids (NEAAs) are synthesized from glucose or other amino acids. In contrast, the essential amino acids (EAAs) cannot be synthesized in cells and must be obtained from external sources. Therefore, the availability of EAAs is the primary limiting factor in cancer cell homeostasis and growth.

Consequently, the research community has been focused on several amino acid transporters, including LAT1, as potential targets for cancer therapeutics. LAT1 is a promising candidate due to its critical role in transporting many essential amino acids, including BCAAs and aromatic amino acids. Studies from multiple groups have also shown that LAT1 is implicated in the occurrence and development of various tumors [17,18,20,21,23]. As such, it is critical to establish the role of LAT1 inhibition for cancer therapeutics. Unexpectedly, one group reported that pharmacological inhibition and genomic mutation of LAT1 does not abolish cell growth or mTORC1 activity in liver cancer cell lines, indicating the possibility of cell-line-specific compensation or a large reserve capacity [32]. However, a closer look into the publication suggests otherwise with reduction in cell growth of Huh7 clone that harbored a successful knockdown of LAT1. Consistently, LAT1 KO is the top candidate suppressing cell proliferation among all the amino acid SLC transporters, as revealed by our analysis based on whole-genome CRISPR KO screening results (provided by DepMap). This suggests a distinctive dependency of HCC progression on large neutral amino acid transport. Taken together, these data necessitate a more detailed and thorough investigation where we established that a complete ablation of LAT1 in Huh7 cells is indeed critical for reducing cell growth and mTORC1 activity.

While LAT1 knockdown arrests cell growth in Huh7 cell line, such effects are less evident in Hep3B and HepG2 cell lines although we observe a lower rate of growth in the CRISPR-knockout cell lines (Appendix A). Notably, the mTOR pathway signaling is downregulated only in the Huh7 cell line (Appendix A). We note that Huh7 possesses a mutant TP53 (Y220C) that is largely stable, Hep3B is p53-null while wild type TP53 is found in HepG2. The higher phosphorylation level of p53 may be able to feedback into the TSC2/MTOR pathway, thus accounting for the differences in signaling activity [33]. Our detailed comparison of HCC cell lines with different TP53 mutations hint the metabolic preference and vulnerability in specific HCC subpopulation and provide a new possibility for precise treatment.

It has also been suggested that inhibition of LAT1 suffers from the redundancy of amino acid transport [34]. The function of LAT1 could be replaced by LAT2 or the combination of LAT3 and TAT1. We found the protein levels of the other members in the LAT family to be unchanged (Figure 2 and Appendix A). The protein levels of the other members in the LAT family for Hep3B and HepG2 cell lines also remained unperturbed. (Appendix A) This indicates that the transcriptional control of the LAT transporters may not be as linked as previously hypothesized [35].

Indeed, the methods employed for the removal of LAT1 may also be important. In this study, we have also utilized shRNA strategy to knockdown LAT1, where no difference was observed in cell proliferation between the knockdown and control cells for all the three cell lines tested (Appendix A). This observation is consistent with a previous report [32]. As such, the extent of the LAT1 ablation may be critical to bring about the essential amino acid deprivation, where LAT1 may become the limiting factor in the rate of amino acid intake.

The results from the paper reveal a promising picture towards developing further therapeutics for LAT1 as a target. Large neutral amino acid deprivation in diet has been discussed as a potential strategy but would require greater investigation to be specific towards cancers whilst not affecting the normal cells [36,37,38]. At the same time, inhibiting mTORC signaling did not present true benefits in clinical practice due to compensation activation of MEK/ERK signaling [39,40]. Blocking the cancer cells selectively of their fuel presents as an attractive strategy to arrest cancer cell growth as evidenced by the cellular and xenograft models employed in this study. Our findings should encourage efforts to develop even better LAT1 inhibitors achieving higher levels of inhibition.

## 4. Materials and Methods

### 4.1. Guide RNA Design and Plasmid Construction

The gene sequences of human *slc7a5* gene were downloaded from NCBI and sgRNAs were designed to target the second exon of the gene using CRISPR design tool (http://crispr.mit.edu, accessed on 12 May 2023). sgRNAs that have BbsI restriction endonuclease site overhangs were synthesized, annealed, and sub-cloned into pSpCas9-2A-GFP plasmids (Addgene #48,138, Cambridge, MA, USA) that is digested with BbsI (NEB, Beverly, MA, USA) using T4 DNA ligase (NEB, Beverly, MA, USA). The resultant constructs were subjected to Sanger sequencing to verify the correct sgRNA sequences.

### 4.2. Cell Culture and sgRNA Transfection

Huh7, Hep3B, and HepG2 cells were maintained in high glucose Dulbecco’s Modified Eagle’s Medium (Gibco; Thermo Fisher Scientific, Inc., Waltham, MA, USA) with 10% Fetal Bovine Serum (Gibco; Thermo Fisher Scientific, Inc., Waltham, MA, USA) and 0.1 mg/mL Penicillin ad Streptomycin (Gibco; Thermo Fisher Scientific, Inc., Waltham, MA, USA). All cells were cultured at 37 °C in humidified atmosphere with 5% CO_2_.

### 4.3. CRISPR/Cas9 Mediated Gene Knockout of slc7a5

Huh7, Hep3B, and HepG2 cells were transfected with plasmids encoded with sgRNA. Plasmids were transfected into 80% confluent cells in 6-well plate (1 cm^2^) plates using JetPRIME (Polyplus, Illkirch, France) transfection reagent according to the manufacturer’s instructions. Briefly, 3 μg of plasmid was diluted into 200 μL of jetPRIME buffer and mixed well before adding 7.5 μL of jetPRIME reagent. This mixture was incubated in room temperature for 10 min and thereafter added dropwise to the cells in serum containing DMEM medium. The medium was changed after 4 h of transfection. 48 h after transfection, cells were subjected to FACS isolation of cells with high levels of GPF using BD FACS Aris III instrument (BD Biosciences, CA, USA). The cells were detached using TrypLE, then single cell suspension was collected by passing through a 50 μm mesh and sorted for a desired GFP fluorescence levelled cells. Non-transfected cells were used for background fluorescence level. Isolated cells were plated in clonal conditions (500 individualized cells in 250 mm plates) and grown for further detection of indels. Each clone was picked and analysed for LAT1 expression by DNA sequencing and confirmed by immunoblotting. Finally, one clone in each cell lines for LAT1 KO were selected for this study.

### 4.4. Cell Proliferation Assay

Cell proliferation of LAT1^WT^ and LAT1^KO^ were monitored for live cell proliferation using the xCELLigence RTCA system (Agilent, Santa Clara, CA, USA), Sulforhodamine B (SRB) assay, and Bromodeoxyuridin (BrdU) incorporation assay. For live cell proliferation, cells were plated in the E-Plates and monitored proliferation every 30 min interval from the time of plating until the end of the experiment. The SRB assay relies on the stoichiometric binding of SRB dye to proteins under mild acidic conditions and its subsequent extraction under basic conditions. The amount of dye extracted is a proxy for cell number in a sample and this assay allows its use for analysis of cell proliferation. For SRB assay, cultured cells were fixed with 10% TCA after an incubation period on plates and stained with SRB for 30 min then excess dye was removed by washing with 1% acetic acid and dried. The bounded SRB is solubilized in 10 mM Tris base solution for optical density measurement at 510 nm using a microplate reader. For BruU assay (Roche, Basel, Switzerland), cells were plated in 96 well plate and incubated for 12 h, then 10 μM of BrdU was added into each cell. After 4 h of further incubation, plasma membranes were permeabilized followed by DNase I digestion. BrdU was detected by anti-BrdU antibody followed by HRP-conjugated secondary antibody incubation. The incorporated BrdU is measured at 450 nm using a microplate reader. In each proliferation assay, cells were plated in quadruplicate per cell line.

### 4.5. BCAA Content Measurement

Cellular BCAA content was measured with Branched Chain Amino Acid Assay Kit (abcam, Waltham, MA, USA) and followed the manufacturer’s protocol. Briefly, Cells were plated into six-well plates in triplicate per cell line. 24 h after seeding, cells were washed twice with pre-warmed HBSS and then pre-incubated with pre-warm HBSS for 2 min. Cells were then incubated with pre-warmed DMEM for 2 h then homogenized and centrifuged to remove cell debris and insoluble materials. Each sample with assay buffer was incubated for 30 min then measured optical density at 450 nm in a microplate reader. BCAA concentrations of test samples were calculated by standard curve (from 0 to 10 nmol) and corrected background by subtracting the value derived from the zero BCAA standards. These measurements were normalized to cellular protein content measured by BCA assay with the remaining lysate of each sample.

### 4.6. Western Blot Analysis

For Western blot analysis, cells were washed with ice-cold PBS twice and then lysed with NP-40 buffer (50 mM Tris-HCl (pH = 7.2), 150 mM NaCl, 1% NP-40, 5 mM EDTA) freshly contained protease inhibitor (Roche, Basel, Switzerland) and phosphatase inhibitor (Roche, Basel, Switzerland). Cell lysates were cleared by centrifugation at 13,000 rpm for 15 min in cold conditions and protein concentration was determined using the BCA protein assay kit (Sigma-Aldrich, St. Louis, MO, USA). Thirty micrograms of cell lysates were separated by SDS-PAGE at the appropriate percentages and transferred onto nitrocellulose membranes. Western blot analysis was conducted following the manufacturer’s instructions and using specific antibodies against LAT1 (#5347, Cell Signaling, Danvers, MA, USA), LAT2 (ab75610, abcam), LAT3 (#HPA018813, Sigma, St. Louis, MO, USA), LAT4 (#HPA021564, Sigma), 4EBP1 (#9644, Cell Signaling), p-4EBP1 (#2855, Cell Singaling), ULK1 (#8054, Cell Signaling), p-ULK1 (#6888, Cell Signaling), mTOR (#2972, Cell Signaling), p-mTOR (#2971, Cell Signaling), p70S6K (#34475, Cell Signaling), p-p70S6K (#9234, Cell Signaling), Raptor (#2280, Cell Signaling), p-Raptor (#2083, Cell Signaling), TSC1 (20988-1-AP, Proteintech, Rosemont, IL, USA), TSC2 (#4308, Cell Signaling), S6RP (#2317, Cell Signaling), p-S6RP (#4858, Cell Signaling), eIF2a (#9722, Cell Sigaling), p-eIF2a (#3398, Cell Signaling), eIF4b (#3592, Cell Signaling), p-eIF4b (#3591, Cell Signaling), GAPDH (sc-32233, Santa Cruz Biotechnology, Dallas, TX, USA). Western blot analysis for mouse liver tissues were conducted using specific antibodies against LAT1 (bs-10125R, Bioss antibodies, Woburn, MA, USA), LAT3 (PA5-37059, Invitrogen, Waltham, MA, USA), LAT4 (PA5-23571, Invitrogen).

### 4.7. Lentivirus Package and Transduction

To construct a lentivirus-mediated overexpression for LAT1, the *slc7a5* gene was amplified by polymerase chain reaction (PCR) with primers designed additional Not1 (n-terminal) and EcoR1 (c-terminal). PCR product was separated, cut, purified then sub-cloned into pLenti-EF1a-PGK-Puro that is digested with Not1 and EcoR1 (NEB, Beverly, MA, USA) using T4 DNA ligase (NEB, Beverly, MA, USA). The resultant constructs were subjected to Sanger sequencing to verify the correct *slc7a5* sequence. The constructed plasmid was co-transfected with pMDLg/pRRE (addgene #12,251), pRSV-rev (Addgene #12,253), and pMD2.G (Addgene, #12,259) into HEK293 cells and the amplified recombinant viruses were isolated via ultracentrifuge. The virus titers were measured with qRT-PCR in AML12 cells. Cells were infected with ten multiplicity of infection of lentivirus and infected cells were selected with puromycin.

### 4.8. Animal Studies and Animal Xenograft Models

All animal procedures were performed by approved protocol by the Institutional Animal Care and Use Committee (IACUC #191480) of the Agency for Science, Technology, and Research (A’STAR) of Singapore. And this study was carried out in strict accordance with the guidelines of the Guide for the Care and Use of Laboratory Animals of the National Institutes of Health and Biological Resource Center IACUC guidelines for cancer research in mice. Cultured LAT1^KO^ clone or LAT1^WT^ control of Huh7 cells were suspended in PBS and mixed with Matrigel (Corning life Science, New York, NY, USA) in 1:1 volume ratio to give a final concentration of 5 × 10^7^ cells/mL. The cell suspension was subcutaneously injected into the lower flank of 8-week-old male CB-17 SCID mice (5 × 10^6^ cells, 0.1 mL/animal). Tumor growth was monitored every three days and the size of tumor was measured by callipers to calculate volumes.

### 4.9. Immunohostochemistry

Immunohistochemistry was performed on 4-μm-thick sections of paraffin-embedded specimens using anti-Ki67 (ab15580, abcam) at 4 degrees in a moist chamber overnight. Then treated with the biotin-labelled secondary antibody, a VECTASTAIN Elite ABC peroxidase kit (PL-6100, Vector laboratories), followed by colour development with DAB. 40× images were acquired with Ni-E brightfield microscope (Nikon, Tokyo, Japan)

### 4.10. Quantitative Real-Time PCR for Gene Expression Analysis

Total RNA from DEN-induced mouse liver tumors and non-tumor liver tissues were extracted using RNeasy mini kit (Qiagen, Hilden, Germany) according to manufacturer’s protocol. 1.2 ug of total RNA was used for reverse transcribed into cDNA by Superscript III reverse transcriptase (Invitrogen, Waltham, MA, USA). Real-time PCR reactions were performed with the Fast Real-Time PCR System (Applied Biosystems, Waltham, MA, USA), using 1 μg of cDNA product in a reaction volume 10 μL. The primers were as follows: *slc7a5* (F: AGCGTCCCATCAAGGTGAAT, R: GGGCTTGTTCTTCCACCAGA), *slc7a8* (F: GCACGAGCGTTTAGAAAAAGACT, R: AGCCAATGATGTTCCCTACAAT), *slc43a1* (F: TTGGAGATGCCAGAGATGGG, R: TCTTGATGAGGCAGGCGATG), *slc43a2* (F: GCCCCATCGTTTATGCACAG, R: GAGGGCAACTGTCTTCTGGTC), *gapdh* (F: AGGTCGGTGTGAACGGATTTG, R: TGTAGACCATGTAGTTGAGGTCA).

### 4.11. Short Hairprin Oligonucleotide Design

Short hairpin oligonucleotides targeting human *slc7a5* gene were subcloned into pRRL lentiviral vector. To generate lentiviral knockdown constructs, the following oligonucleotides were used for vector construction: *slc7a5*-1: TGCTGTTGACAGTGAGCGAAAGGGTGATGTGTCCAATCTATAGTGAAGCCACAGATGTATAGATTGGACACATCACCCTTCTGCCTACTGCCTCGGA; *slc7a5*-2: TGCTGTTGACAGTGAGCGATGGAAAGTAGCTGCTAGTGAATAGTGAAGCCACAGATGTATTCACTAGCAGCTACTTTCCACTGCCTACTGCCTCGGA; *slc7a5*-3: TGCTGTTGACAGTGAGCGCTCCCTCCTTTGTTTACTTATATAGTGAAGCCACAGATGTATATAAGTAAACAAAGGAGGGAATGCCTACTGCCTCGGA. The sequence of non-targeting control (NTC) is as follow: CCTAAGGTTAAGTCGCCCTCG.

### 4.12. Public Data Availability and Processing

Transcriptomics and proteomics data of HCC patients were acquired from the Cancer Genome Atlas (TCGA) and Clinical Proteome Tumor Analysis Consortium (CPTAC), respectively. Dependency scores of CRISPR KO screening were downloaded by R package *depmap* from the Cancer Dependency Map (DepMap) databases [41]. All the SLC transporters for amino acids were ranked by average dependency scores.

### 4.13. RNA Sequencing Analysis

Total RNA was extracted from Huh7 cell lines with TRIzol Reagent according to the manufacturer’s instructions. RNA samples were treated with DNase I (Thermo Fisher Scientific, Inc., Waltham, MA, USA) and cleaned with the Ribo-Zero Magnetic Core Kit (Illumina, San Diego, CA, USA). A total of 500 ng RNA was used for library preparation with TruSeq Stranded Total RNA with Ribo-Zero Gold kit (Illumina, RS-123-2201). Cutadapt and RSeQC were executed for adaptor trimming and quality control of paired-end RNA sequencing data then aligned to the UCSC hg19 human reference genome with TopHat2.0 [42]. Read-pairs were counted with HTSeq, and differential gene expression were determined by DESeq2 package [43]. Genes with |log2(fold of changes)| ≥ 1 and Benjamini–Hochberg (BH) adjusted *p*-values < 0.05 were defined as differentially expressed genes (DEGs). Transcripts per kilobase million (TPM) of DEGs were used for hierarchical clustering and heat map visualization. Pathway analysis were performed by Ingenuity Pathway Analysis (IPA) [44].

### 4.14. Survival Analysis

Survival analysis were performed by ToPP [45] based on TCGA-LIHC dataset. The optimal cutoff is defined as the point with the most significant split in log-rank test by R package *survival*.

## Figures and Tables

**Figure 1 ijms-24-09171-f001:**
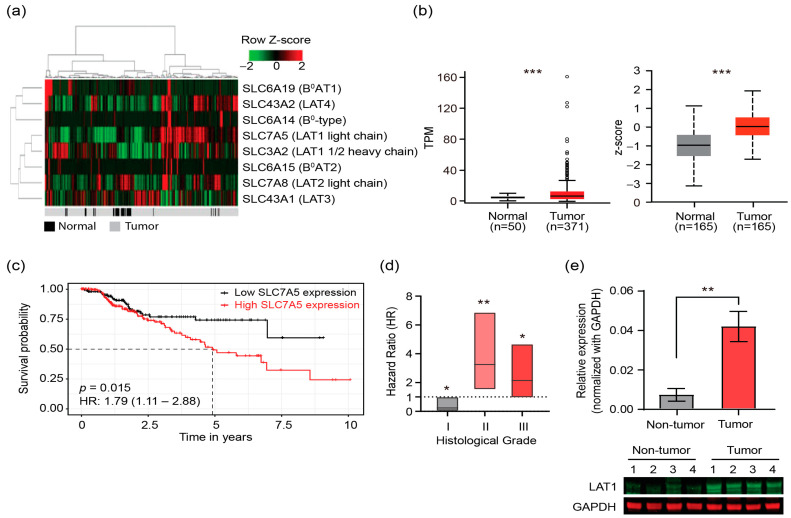
Expression of LAT1 is associated with clinical outcomes of HCC patients. (**a**) Heatmap of normalized expression of various BCAA transporters (rows) in all HCC samples (columns) from TCGA data. Both rows and columns are clustered by Pearson correlation, and gray-black bar (bottom) shows cancer status of samples. (**b**) mRNA expression (in TPM) and protein expression (expressed as z-score) of LAT1 in normal liver tissues (gray) and HCC (red). (**c**) Kaplan-Meier analysis of disease-specific survival (DSS) for TCGA-LIHC patients according to the mRNA expression level. (**d**) Hazard ratio (HR) of DSS in the patients with grade I-III HCC tumors according to the LAT1 expression levels. Patients were classified into two groups, i.e., patients with high LAT1 expression and those with low expression according to the optimal cut off. (**e**) qRT-PCR analysis and Western blot analysis of LAT1 in DEN mouse model. The qRT-PCR data is average of three independent experiments (*n* = 3) ± S.E.M. *, *p* < 0.05; **, *p* < 0.01; ***, *p* < 0.005.

**Figure 2 ijms-24-09171-f002:**
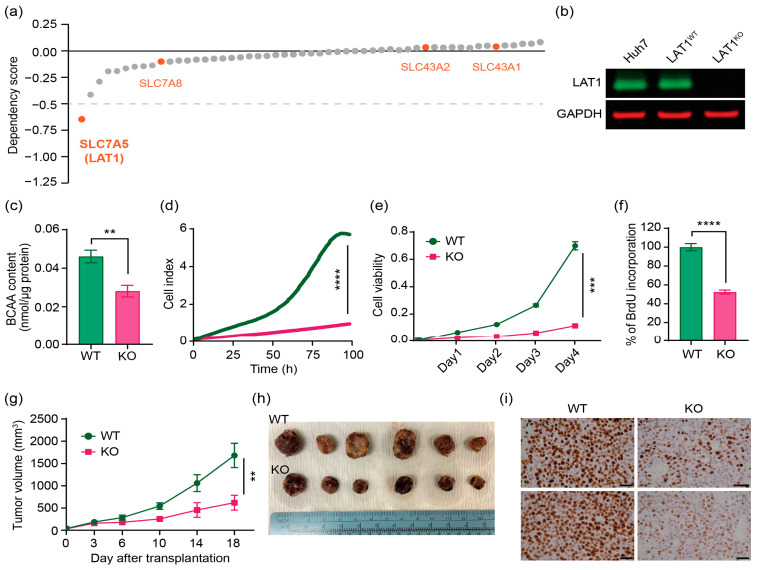
CRISPR KO of LAT1 results in lower HCC cell proliferation in cell line and xenograft models. (**a**) Average dependency score of amino acid transporters for the proliferation/survival of HCC cell lines based on DepMap CRISPR KO screening data. The four LAT isoforms are labeled in orange. (**b**) Western blot analysis of cell lysates shows no detectable LAT1 protein in the LAT1^KO^. LAT1 is completely knocked-out in LAT1^KO^, while expressed in LAT^WT^. (**c**) Intracellular BCAA content was measured in LAT1^WT^ and LAT1^KO^ samples. Real-time proliferation analysis (**d**), colorimetric SRB analysis (**e**), and BrdU incorporation assay (**f**) were performed to evaluate proliferation rates in the LAT1^WT^ and LAT1^KO^ clones. The data represented are average of three independent experiments (*n* = 3) ± SEM. (**g**) Tumor growth curves of subcutaneous xenograft tumor model mice. Tumor volumes (in mm^3^) were calculated as Length x (Square of Width)/2. (**h**) Representative images of xenograft tumors after three weeks of injection. (**i**) Immunohistochemistry of Ki67 staining in xenografts generated from subcutaneous transplantation of LAT1^WT^ and LAT1^KO^ of Huh7 cells into mice. Scale bar = 100 μm. The data represented are average of three independent experiments (*n* = 3) ± SD. Comparisons among groups were analyzed with two-way ANOVA and between two groups with Student’s *t* test. **, *p* < 0.01; ***, *p* < 0.005; ****, *p* < 0.001.

**Figure 3 ijms-24-09171-f003:**
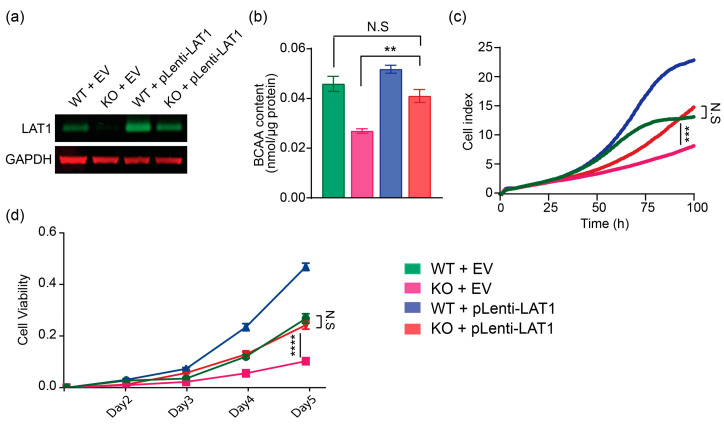
Rescue of proliferation defect in LAT1^KO^ by LAT1 re-expression. (**a**) Western blots showing upregulation of LAT1 after lentiviral infection of LAT1^KO^. (**b**) LAT1 transport activity of LAT1^WT^ and LAT1^KO^ stably expressing either EV or pLenti-LAT1 was determined by BCAA assay measurement. The data represented are average of three independent experiments (*n* = 3) ± SEM. **, *p* < 0.01. (**c**) Real-time proliferation analysis and (**d**) SRB assay (with optical density at 510nm as readout) of LAT1^WT^ and LAT1^KO^ stably expressing either EV or pLenti-LAT1 show that LAT1 rescues proliferation defect in LAT1^KO^. The data represented are average of three independent experiments (*n* = 3) ± SD. Comparisons among groups were analyzed with two-way ANOVA and between two groups with Student’s *t* test. **, *p* < 0.01; ***, *p* < 0.005; ****, *p* < 0.001.

**Figure 4 ijms-24-09171-f004:**
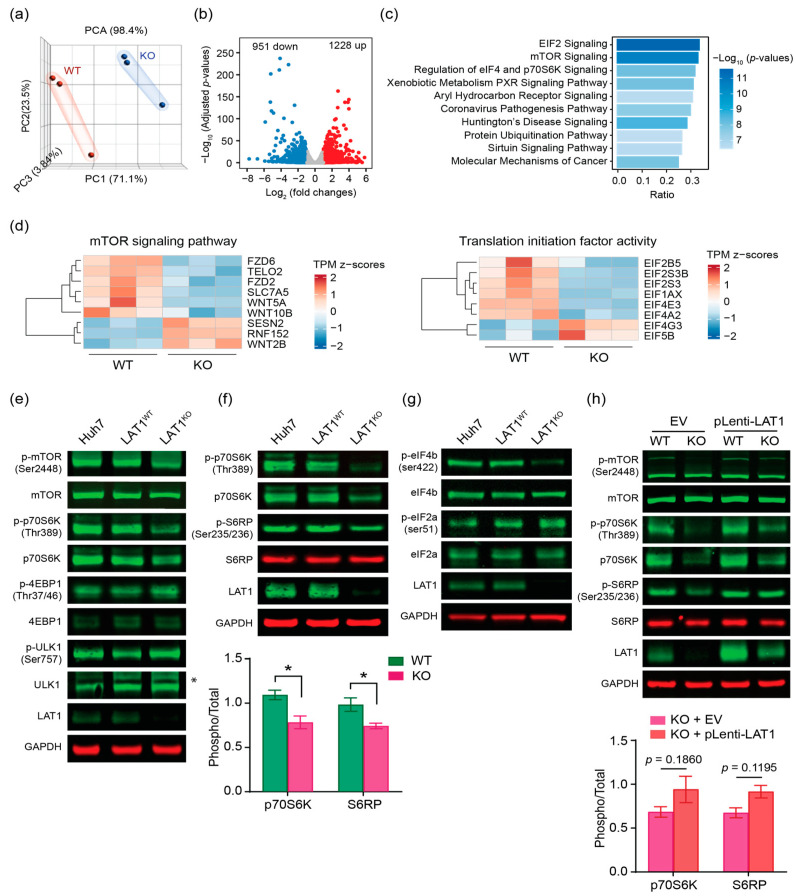
LAT1 knockout decreases mTORC1 activity. (**a**) PCA result for RNA-seq data of cultured LAT1^KO^ and LAT1^WT^ cells. The figure displays a three-dimensional scatter plot and samples with similar gene expression profiles are clustered together. (**b**) Volcano plot showing significantly altered genes between LAT1^KO^ and LAT1^WT^. Red and blue colors indicate the genes with significantly increased and decreased expression, respectively. (**c**) Top 10 significantly enriched pathways identified by IPA using the expression profile of in the LAT1^KO^ cells. The horizontal bars indicate the different pathways with highest gene ratio and P values less than 0.05. (**d**) Heat map shows the DEGs of mTOR signaling and translation initiation factor activity with |log fold change|≥ 1.0 between two groups. (**e**) Control Huh7 cell, LAT1^WT^, and LAT1^KO^ were cultured for 24 h in normal DMEM then changes in mTORC1 signaling were assessed by immunoblotting of phosphorylation and total MTOR, 4EBP1, p70S6K, ULK1. (**f**) Phosphorylation and total p70S6K and S6RP of control Huh7 cell, LAT1^WT^, and LAT1^KO^ lysates were analysed by immunoblotting. The means of p-p70S6K/p70S6K and p-S6RP/S6RP are expressed as percentages compared with LAT1^WT^ and LAT1^KO^. The data shown are averages of four independent experiments performed (*n* = 4) ± S.D. *, *p* < 0.05. (**g**) Phosphorylation and total eIF4b and eIF2a of control Huh7 cell, LAT1^WT^, and LAT1^KO^ lysates were analysed by immunoblotting. (**h**) Phosphorylation and total mTOR, p70S6K, and S6RP of LAT1^WT^ and LAT1^KO^ transduced by either EV or pLenti-LAT1 were analysed by immunoblotting. The means of p-p70S6K/p70S6K and p-S6RP/S6RP are expressed as percentages compared with LAT1^KO^ transduced with EV or pLenti-LAT1. The data represented are averages of two independent experiments performed (*n* = 2) ± S.D. GAPDH was used as a loading control for Western blot analysis.

## Data Availability

We will be uploading the raw RNA seq data to GeoDataSets as soon as the manuscript is accepted.

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
