# Peer review of "Genetic Ablation of LAT1 Inhibits Growth of Liver Cancer Cells and Downregulates mTORC1 Signaling"

_ijms, 2023, doi:10.3390/ijms24119171_

Round 1
Reviewer 1 Report
This is an extensive study with a huge data. I found it appropriate for publication in the IJMS with a minor modification. The paper is well written and the topic is appropriate. The aim of the manuscript is well described and the results and discussion were well prepared, its results and discussion section are correlated to the cited literature data.
I suggest providing a schematic overview of this study to better understand the readers.
Moderate English checking required.
Author Response
We thank Reviewer 1 for the comments. We have added a graphical abstract as suggested by the Reviewer to allow readers better understand the study.
Reviewer 2 Report
I read and reviewed the paper entitled "Genetic ablation of LAT1 inhibits growth of liver cancer cells 2 and downregulates mTORC1 signaling" by kim et al. This is a translation study investigating the effects of the aminoacid transporter LAT1 to liver cancer cell proliferation. The paper is interesting and the authors used different methodologies and models to prove the growth dependency of HCC cells to LAT1.
I have found minor issues that should be checked before publication.
1. Figure 1, LAT1 western blot from human specimen should be replaced with a more comprehensible blot. I checked the antibody that was used and I think the cut you show is a higher than the predicted MW of the protein.
2. The abstract is not easy to read, I advise to write a structured abstract and to limit the background that might be better stated in the introduction section.
Author Response
I have found minor issues that should be checked before publication.
- Figure 1, LAT1 western blot from human specimen should be replaced with a more comprehensible blot. I checked the antibody that was used and I think the cut you show is a higher than the predicted MW of the protein.
Response: We thank the Reviewer for the comment. In Figure 1E, the Western blot is probed for LAT1 in the DEN mouse model. For the mouse specimen, we have used LAT1 antibody specifically for mouse, that is Cat No: bs-10125R from Bioss antibodies. In the company website, it could be seen that the LAT1 band is expected to be at around 55 kDa. As such, we take reference with the band above 50kDa in our Western blot for LAT1.
2. The abstract is not easy to read, I advise to write a structured abstract and to limit the background that might be better stated in the introduction section.
In order to allow readers to better understand our paper and the abstract, we have removed the first two sentences and the abstract is more direct. In addition, we have added a graphical abstract in the latest submission.
